# Dysregulation of DNA Methylation and Epigenetic Clocks in Prostate Cancer among Puerto Rican Men

**DOI:** 10.3390/biom12010002

**Published:** 2021-12-21

**Authors:** Anders Berglund, Jaime Matta, Jarline Encarnación-Medina, Carmen Ortiz-Sanchéz, Julie Dutil, Raymond Linares, Joshua Marcial, Caren Abreu-Takemura, Natasha Moreno, Ryan Putney, Ratna Chakrabarti, Hui-Yi Lin, Kosj Yamoah, Carlos Diaz Osterman, Liang Wang, Jasreman Dhillon, Youngchul Kim, Seung Joon Kim, Gilberto Ruiz-Deya, Jong Y. Park

**Affiliations:** 1Department of Biostatistics and Bioinformatics, H. Lee Moffitt Cancer Center, Tampa, FL 33612, USA; anders.berglund@moffitt.org (A.B.); ryan.putney@moffitt.org (R.P.); youngchul.kim@moffitt.org (Y.K.); 2Department of Basic Sciences, Ponce Research Institute, School of Medicine, Ponce Health Sciences University, Ponce 00716-2347, Puerto Rico; jmatta@psm.edu (J.M.); jencarnacion@psm.edu (J.E.-M.); carmenortiz@psm.edu (C.O.-S.); jdutil@psm.edu (J.D.); rlinares20@stu.psm.edu (R.L.); jmarcial20@stu.psm.edu (J.M.); cabreu20@stu.psm.edu (C.A.-T.); cjdiaz@psm.edu (C.D.O.); gruiz@psm.edu (G.R.-D.); 3Burnett School of Biomedical Sciences, University of Central Florida, Orlando, FL 32816, USA; Ratna.Chakrabarti@ucf.edu; 4Biostatistics Program, School of Public Health, Louisiana State University Health Sciences Center, New Orleans, LA 70112, USA; hlin1@lsuhsc.edu; 5Department of Radiation Oncology, H. Lee Moffitt Cancer Center, Tampa, FL 33612, USA; Kosj.Yamoah@moffitt.org; 6Department of Molecular Biology, H. Lee Moffitt Cancer Center, Tampa, FL 33612, USA; Liang.wang@moffitt.org; 7Department of Pathology, H. Lee Moffitt Cancer Center, Tampa, FL 33612, USA; Jasreman.Dhillon@moffitt.org; 8Department of Internal Medicine, Catholic University of Korea, Seoul 06591, Korea; cmcksj@catholic.ac.kr; 9Department of Cancer Epidemiology, H. Lee Moffitt Cancer Center, Tampa, FL 33612, USA

**Keywords:** prostate cancer, DNA methylation, aggressiveness, Hispanic/Latino population, ancestry structure

## Abstract

In 2021, approximately 248,530 new prostate cancer (PCa) cases are estimated in the United States. Hispanic/Latinos (H/L) are the second largest racial/ethnic group in the US. The objective of this study was to assess DNA methylation patterns between aggressive and indolent PCa along with ancestry proportions in 49 H/L men from Puerto Rico (PR). Prostate tumors were classified as aggressive (*n* = 17) and indolent (*n* = 32) based on the Gleason score. Genomic DNA samples were extracted by macro-dissection. DNA methylation patterns were assessed using the Illumina EPIC DNA methylation platform. We used ADMIXTURE to estimate global ancestry proportions. We identified 892 differentially methylated genes in prostate tumor tissues as compared with normal tissues. Based on an epigenetic clock model, we observed that the total number of stem cell divisions (TNSC) and stem cell division rate (SCDR) were significantly higher in tumor than adjacent normal tissues. Regarding PCa aggressiveness, 141 differentially methylated genes were identified. Ancestry proportions of PR men were estimated as African, European, and Indigenous American; these were 24.1%, 64.2%, and 11.7%, respectively. The identification of DNA methylation profiles associated with risk and aggressiveness of PCa in PR H/L men will shed light on potential mechanisms contributing to PCa disparities in PR population.

## 1. Introduction

Based on American Cancer Society data, 248,530 new prostate cancer (PCa) cases and 34,130 PCa-specific deaths are anticipated in the US in 2021 [1]. The lifetime risk of PCa is 12.5% [2]. PCa-specific mortality (PCSM) rates have been found to vary among different ethnic groups in the US. The study by Chinea et al. (2017) reported differences in PCSM rates when comparing Hispanic/Latino (H/L) subgroups to non-Hispanic Whites (NHWs) and non-Hispanic Blacks (NHBs) [3]. However, this study aggregated all H/L subgroups into one broad group including: Mexican Americans, Puerto Ricans, Cubans, South or Central Americans, and Dominicans. Therefore, we need to investigate variations in H/L subgroup specific PCSM rates. Among different H/L subgroups, Puerto Rican (PR) men showed much higher PCSM rates than other Hispanic groups and NHBs [3]. Indeed, in Puerto Rico (PR), PCa is the most common type of cancer case and accounts for the most cancer-specific deaths [4].

PCa is a complex disease that is mediated by the accumulation of genetic and epigenetic aberrations, such as altered androgen receptor activity, changes in chromatin structure, differential expression of oncogenes and tumor suppressor genes, or defective cell division [5]. Differential DNA methylation can influence carcinogenesis and disease progression [6]. Indeed, the most common molecular event in PCa is dysregulation of DNA methylation. Among these epigenetic changes, some specific changes may be associated with poor outcomes, including PCSM, metastasis, and recurrence [7]. A study from the Cancer Genome Atlas (TCGA) found associations between gene expression and methylation profiles. This study suggested that epigenetic changes define distinct molecular subtypes of PCa [8]. The role of DNA methylation in promoter regions has been investigated many times, and most hypermethylation has been related with gene silencing of tumor suppressor genes in PCa and with poor outcomes [6,9,10,11]. PCa is typically known as a slowly developing disease. However, approximately 20% of cases are classified as aggressive. The aggressive PCa phenotype is associated with the development of metastasis and poor survival outcomes.

Mateo et al. (2015) reported that the dysregulation of tumor suppressor genes are often found in aggressive PCa [5]. Promoter hypermethylation may drive cancer through tumor suppressor gene inactivation and activation of oncogenes [12]. Therefore, additional studies are needed to further understand the epigenetic regulation of tumor suppressor genes and oncogenes in PCa.

Numerous studies reported that differential DNA methylation influences the likelihood of developing PCa and also affects its progression [13,14]. Yang and Park (2012) conducted an extensive review of over 100 studies and presented a list of frequently reported differentially methylated genes in malignant prostate tissue [14]. Most of the studies reviewed investigated a small number of genes because of a small sample size in order to increase the chance to yield statistically significant results. However, this approach may not assess methylation impact associated with multiple genes. Several of the studies used an epigenome-wide methylation approach, often used to examine multiple genes. As expected, numerous differential methylated genes were identified. However, another data set is needed to validate findings [15,16,17]. Since differential DNA methylation may influence health disparities in PCa [18], there is a need to investigate methylation profiles to evaluate potential PR-specific methylated genes.

Aging is one of the major risk factors for PCa. The risk of carcinogenesis in any given tissue is closely related with the mitotic age of the tissue and therefore the cumulative number of cell divisions [19]. The turnover rate of tissues is affected by several factors, such as inflammation, injury, and exposure to carcinogens [20]. Therefore, increased turnover in tissues may increase molecular alterations and eventually lead to carcinogenesis [21]. DNA methylation may be involved in aging [22]. DNA methylation biomarkers for aging, also known as the epigenetic clock, have been developed based on DNA methylation data. DNA methylation age (DNAmAge), generated from the epigenetic clock, estimates epigenetic age, measures subject age, and, more importantly, predicts disease risk [23]. This DNAmAge can be used for predicting chronic diseases, including cancers, if these values show substantial deviations [24]. With this model, we can estimate epigenetic age and predict cancer risk [25]. In this study, the total number of stem cell divisions (TNSC) and stem cell division rate (SCDR) were compared between tumor and adjacent normal tissues using this method [25].

Currently, the dysregulation of DNA methylation in PR PCa patients is well known. We recently reported differentially methylated genes in PCa tumor tissues and methylated genes associated with aggressiveness in a small number of PR men with PCa [26]. Our findings on epigenetic differences between prostate tumor and adjacent non-involved (normal) tissues will be used to detect PCa early and may clarify how PCa starts from the normal tissues. In addition, our goal is to identify PR specific DNA methylation biomarkers responsible for cancer disparities in PR H/L men. In addition, we characterized the ancestry structure of PR patients who participated in this study. Via et al. analyzed 642 PR individuals for ancestry structure and reported that PR is an admixed population [27].

## 2. Materials and Methods

### 2.1. IRB Approval, and Tissue Sample Selection

Two Institutional Review Boards approved this study: the Moffitt Cancer Center (Protocol no. Pro00048100) and the Ponce Health Sciences University (PHSU) (Protocol no. 1909021277A001). All study participants signed an Informed Consent. We obtained 49 formalin-fixed paraffin-embedded (FFPE) prostate tumor and adjacent non-involved pair samples from the Puerto Rico Biobank (PRBB), a U54 PHSU-MCC PACHE Partnership core facility. Based on Gleason scores, tumors from study participants were classified as either aggressive, 17, or as indolent, 32.

### 2.2. DNA Methylation Analysis

#### 2.2.1. Illumina EPIC Methylation and DNA Samples

We used the Illumina Infinium Methylation EPIC (EPIC) BeadChip DNA methylation platform to obtain DNA methylation levels in DNA samples from FFPE-preserved tissues as described in the manufacturer’s instructions. This instrument is located at the Molecular Genomics Core, MCC, Tampa, FL, USA. Genomic DNA was obtained from the prostate tissues. DNA was extracted from the marked tumor area on the H&E slides by the pathologist (J.D.). DNA quality was tested with DNA integrity numbers (DINs).

#### 2.2.2. Quality Control and Normalization for Data Obtained from Epigenome-Wide Methylation Assays

Raw IDAT files were read by the minfi (version 1.28.4) [28,29] Bioconductor package for R (version 3.5.2). Minfi’s implementation detection *p*-value was used to calculate detection *p*-values. Normalization was performed using the normal-exponential out-of-band (NOOB) [30] method. Next, functional normalization (FunNorm) [31], a between-array normalization method, was used. As Illumina recommended, the preprocess Funnorm function returned an object containing β-values which were measured with an offset of 100 in the denominator [32]. To visualize data quality and identify outliers and potential batch effects, we used a histogram of β-values, the number of missing values, and principal component analysis (PCA). We set β-values with a corresponding detection *p*-value > 0.05 as missing values.

#### 2.2.3. Detection of Differentially Methylated Regions (DMRs)

Student’s *t*-test and false discovery rates (FDR) were used for two group comparisons and multiple testing respectively [33]. We set the minimum point for the mean β-value between the two groups as 0.2. Therefore, CpG sites were considered significant if the difference in for tumor versus normal was >0.2 and 0.05 when comparing low-risk (indolent) and high-risk (aggressive) [34]. We used a region-based analysis to prevent potential false positives. We considered all the CpG probes within a specific gene and determined as DMRs if several CpG probes within that gene were differentially methylated [35,36,37,38]. We did not consider as a DMR if only one CpG probe was different within a specific gene. Statistical analysis was performed using MATLAB (Natick, MA, USA).

### 2.3. Analysis for Ancestry Structure

The ancestry informative markers (AIMs) were used to estimate ancestry structure of the 49 Puerto Rican men with the prostate cancer previously described. AIMs, 106 genetic variations, provide the proportion of indigenous American, African, and European ancestry. Genotyping was performed by a multiplex PCR coupled with single-base extension methodology using a Sequenom analyzer. We excluded 5 single nucleotide polymorphisms because of weak clustering or less than optimal genotyping rates (<90%). Admixture [39] was used to estimate ancestry structure.

### 2.4. Estimation of Epigenetic Mitotic Clocks

The total number of stem cell divisions (TNSC) and stem cell division rate (SCDR) were calculated using epiTOC2 methods as described previously [25].

## 3. Results

### 3.1. Demographic and Clinicopathological Characteristics of Study Group

The mean age at diagnosis for PR H/L men in the two groups of PCa was 64.6 years for aggressive (high-risk) group, and 60.3 years for the indolent (low-risk) group. Thirty-five percent of all patients (*n* = 17) had a high Gleason score (7 (4 + 3) or 8 − 10) and were classified as a high-risk group. As expected, the significantly different distribution in clinical stage was detected between two groups. No statistically significant differences (*p* > 0.05) between the two groups were evident in terms of prostate-specific antigen (PSA) levels and surgical margins (Table 1).

### 3.2. Differential Methylation between Prostate Cancer Tumors and Adjacent Non-Involved Tissue

Quality control analysis, PCA, β-value distributions, and missing values identified and excluded three samples (two tumors and one adjacent non-involved) which did not meet quality control criteria. The first principal component (PC1) showed a clear separation between tumor (*n* = 49) and adjacent non-involved tissues (*n* = 49) from the unsupervised PCA model based on the remaining DNA samples (*n* = 98) (Figure 1A). These data suggested significant differential methylation between adjacent non-involved and tumor tissues. To measure differential methylation levels between adjacent non-involved and tumor tissues, a two-group comparison was performed with a volcano plot. This is a type of scatterplot that shows statistical significance (*q*-value) versus magnitude of change (Δ*β*-value). In this figure, the most hypermethylated CpG probes are towards the right, the most hypomethylated CpG probes are towards the left, and the most statistically significant genes are towards the top. A volcano plot was based on the false discovery rate (FDR)-corrected *p*-values (*q* < 0.001) and the mean difference (Δ*β*-value > 0.2) between the two groups (Figure 1B). From this analysis, we identified 8293 differentially methylated CpG probes; the majority of probes (*n* = 7744) were hypermethylated in tumor samples (Figure 1B). A scatter density graph of the average *β* value for tumor vs. adjacent normal samples demonstrated that many CpG-probes have a similar methylation level in tumor and normal samples, since most of the CpG-probes are along the diagonal (Figure 1C). The “bump” located at *x* = 0.24, *y* = 0.5 indicates an increase of methylation in PCa tumor samples. A histogram of the Δ*β*-value showed more hyper-methylated probes than hypo-methylated probes, especially for Δ*β*-value > 0.2 (Figure 1D).

With the DAVID Functional Annotation Tools [40], we grouped genes based on functional similarity in order to better interpret the large lists of genes derived from epigenome-wide analysis. From 12 functional annotation categories, the highest enriched gene-set for hyper-methylated probes was the Homeobox group (Figure 1E). We compared the Δ*β*-value for significant hyper-methylated CpG-probes based on functional locations, DNase Hypersensitivity CpG-probes (DHS), Open Chromatin probes (OC), and Transcription factor binding sites (TFBS) (Figure 1F).

Based on DMR selection criteria, we identified 3034 differential probes in 892 genes. Some identified genes, such *GSTP1* (Figure 2A), *RARB* (Figure 2B), and *RASSF1* (Figure 2C), which were previously suggested as methylation biomarkers for PCa [14], showed DMRs in tumor tissues. In addition, some genes, including Tumor Protein D52 (TPD52), which is an oncogene [41], showed hypomethylation in multiple CpG sites in prostate tumor tissues (Figure 2D).

Hypermethylation of *RARB* showed a significant association with PCa risk (OR 1.76, 95% CI: 1.29–2.40), and the association was more evident in NHBs (OR 2.18, 95% CI: 1.39–3.44) [40]. A role of methylation in *RASSF1A* gene in PCa risk was reported in a meta-analysis. The odds ratio (OR) of *RASSF1A* methylation in men with PCa, compared to controls, was 14.7 (95% CI = 7.6–28.6), with high specificity (AUC: 0.87, 95% CI: 0.72–0.94) and sensitivity (AUC: 0.76, 95% CI: 0.55–0.89). Therefore, promoter methylation of the *RASSF1A* gene has been suggested as a potential biomarker for PCa risk [41]. Additionally, the hypomethylated genes in this study, such as tumor protein D52 (*TPD52*), an oncogene, have been reported to be over-expressed in PCa as compared with adjacent normal tissues. Therefore, methylation in *TPD52* was proposed as a biomarker for PCa risk [42]. These results can be further evaluated for their contribution to PCa risk in PR PCa patients.

### 3.3. Comparison of Epigenetic Clocks between Tumor vs. Adjacent Normal Tissues

Risk of mutations in a cell are increased by a high turnover rate, and this is associated with an accelerated cell division as part of the cell cycle [21]. Therefore, the mitotic age is correlated with the risk of carcinogenesis [19]. With Tischendorf’s epigenetic mitotic clocks [42], we estimated a mathematical expression to estimate the fraction of cells methylated at 163 candidate CpG sites in tumor or adjacent normal tissues. The total number of stem cell divisions (TNSC) at the patient’s age, and the parameters for prostate-specific probability of de novo methylation and baseline methylation (i.e., at fetal stage) were determined. We note that TNSC and stem cell division rate (SCDR) in tumor tissues were significantly higher than adjacent normal tissues (*p* < 0.0001) (Figure 2E,F).

### 3.4. Differentially Methylated Genes Associated with Aggressive Type of Prostate Cancer

To further investigate the differentially methylated genes in prostate tumor samples, we calculated a PCA model using only the tumor samples. The unsupervised PCA model did not show a clear distinction between tumor and adjacent non-involved tissues in the first principal component (PC1) (Figure 3A).

The *p* values (*p* < 0.001) and the mean difference (Δ*β*-value > 0.05) between the indo-lent and aggressive groups are presented in a volcano plot (Figure 3B). This analysis resulted in 181 differentially methylated CpG probes, with a majority of the probes showing hypomethylation in tumor samples (Figure 3B). The volcano plot shows the significant differentially methylated CpG sites between aggressive and indolent prostate tumors found in 141 genes (Figure 3B). Appendix A presents the gene symbol, *p*-values, probeID, and mean difference of methylation levels in high- and low-risk groups. Some identified genes include RIN2 (Figure 3D), which was previously suggested as a methylation biomarker for early stage of esophageal cancer [43], and MGC29506 (Figure 3C), which was previously suggested as a biomarker for gastric [44] and testicular cancers [23].

Among 181 differently methylated CpG sites, methylation level was significantly increased in 130 sites and significantly decreased in 51 sites, in 141 unique genes (*p* < 0.001), and the strongest evidence was found for *SPRED2* [43], *HLA-C* [44], *TMEM108* [45], and *PRKAG2* [46], which are involved in MARK signaling, immune pathway, and cellular homeostasis.

### 3.5. Ancestry Analysis

We determined the ancestry structure for the 49 study participants. The contribution of African ancestry ranged between < 1% and 85.3%, averaging 21.9% (standard deviation, SD 17.8%) (Figure 4 and Table 2). We found that the European and indigenous American ancestry components were 65.8% and 12.3%, on average, respectively. Interestingly, in Puerto Rican men in this study, there were large variations in European and African ancestries. However, the Indigenous American ancestries were relatively homogenous (Table 2, Figure 4).

## 4. Discussion

Puerto Rican Hispanic/Latino men are at an increased risk of prostate cancer-specific mortality compared with NHW men. However, the relationship between this observation and epigenetics and how this relationship explains PCa racial and ethnic health disparities are controversial [26]. This study represents an ongoing effort to investigate DNA methylation in PR H/L men with PCa. We observed that several differentially methylated genes were found in aggressive tumor tissue from PR PCa patients. Our findings will help us to both understand why and know when PR patients have an aggressive type of prostate cancer. Once we identify these unique DNA methylation patterns at the time the patient is diagnosed, long-term, they may provide new molecular tools to clinicians to determine their treatments for PR H/L men with PCa with high risk.

This study represents the first steps towards implementing personalized medicine in PR PCa patients. This approach is tailored to the biology of the individual, making it more effective than a “one size fits all” approach. It may provide a molecular tool to clinicians to reduce cancer health disparities.

We identified 892 genes that showed significant differential methylation for multiple probes in the prostate tumor tissue. The promising methylation marker candidates were identified in many genes including: *GSTP1, RARB*, *RASSF1,* and *TPD52* (Figure 2). *GSTP1* is one of the metabolizing enzymes and plays a key role in preventing the carcinogenesis caused by environmental exposures [47]. Hypermethylation of the *GSTP1*, a tumor suppressor gene, frequently occurs in different cancer types, including PCa [48]. Hypermethylation of the *GSTP1* is associated with down-regulation of GSTP1 expression, and eventually leads to carcinogenesis. Therefore, hypermethylation of *GSTP1* was suggested as a biomarker of early stage of PCa [49]. Interestingly, *GSTP1* methylation in tumors is more strongly related to NHB men with PCa [50] than NHWs. *RARB* is one of the well-known tumor suppressor genes, frequently hypermethylated in prostate tumorigenesis [51].

Massie et al. (2017) reviewed associations with DNA methylations and PCa and reported a list of frequently differentially methylated genes in prostate tumor tissues [7]. After an extensive review of 17 studies, they identified 5962 differentially methylated genes in tumors. Among 892 genes we identified, 526 (59.0%) genes, including GSTP1 and SMAD3, were validated in the list of genes from Massie’s review.

Methylated genes (*n* = 366) identified in this study were not found in the list of Massie’s study. This may be partly due to differences in methodology, platforms used, sample size, and the unique ancestral composition of the Puerto Rican population, among others, or can be specific to the PR patients studied. These genes observed in this study may contribute to differences in PCa outcomes that have been reported between PR and NHW men. DNA methylation continues to change during progression of PCa with different characteristics at different stages if the disease. Therefore, it is crucial to study the epigenetic clock or DNA methylation age (DNAmAge) to understand a picture of the epigenetic landscape in PCa progression.

Several groups have reported different methylation patterns between PCa tumor and adjacent normal tissues [52]. Using epigenome-wide DNA methylation data derived from 73 PCa tumors and 63 adjacent prostatic tissue samples, Kirby et al. (2017) identified methylation patterns that distinguished prostate tumor from adjacent normal tissue with a high predictive power [53]. However, this study only compared tumors with adjacent normal tissues, and different methylation pattens in aggressive PCa were not assessed. We identified 141 differentially methylated genes, including *MGC29506* and *RIN2*, after comparison of PR patients with aggressive vs. indolent PCa (Appendix A).

Currently, the exact function of the *MGC29506* gene is unknown. Katoh and Katoh (2003) reported that the expression of MGC29506 was down-regulated in intestinal-type gastric cancer as compared with adjacent normal tissues [54]. Further, MGC29506 protein inhibited proliferation of cells by arresting cells at the G0/G1 and S phases of the cell cycle. These results suggested the potential of *MGC29506* as a suppressor gene in gastric cancer [55].

Although *RIN2* has not been studied in terms of its role in PCa, as one of DNA damage and repair related gene sets, this gene has been investigated in colorectal [56] and esophageal [57] cancers. Recently Wang et al. (2021) reported a 12-gene-based prognostic signature selected from 160 DNA damage and repair related genes. This signature can predict survival of patients with colorectal cancer (AUC: 0.80) [56]. Alvi et al. (2013) reported that methylation status of four genes, including *RIN2*, can distinguish between esophageal tumor and benign tissues with high accuracy (AUC: 0.98) [57].

We identified several genes associated with prostate cancer. One of genes we identified is *PRDM16*. Few studies have reported for the role of *PRDM16* in PCa. Chandrashekar et al. reported that PRDM16 expression was associated with the survival of PCa patients with different Gleason scores. Its expression in prostate tumor indicated a high diagnostic value for early detection of PCa [58]. Another gene, TP73 were extensively investigated in prostate cancer. We previously reported a significant role of genetic variation in TP73 in prostate cancer. We detected a significant inverse relationship between p73 variation and PCa aggressiveness. Additionally, p73 variation is marginally associated with overall death as well as PCa-specific death [59].

*SPRED2* is known as one of the key negative regulators of the MAPK signaling pathway. Kachroo et al. observed the downregulation of *SPRED2* in an aggressive type of prostate cancer and *SPRED2* overexpression suppressed prostate cancer cell proliferation [43]. Regulation of HLA expression on cell surface is involved in natural killer cell-medicated lysis of tumor cells [44]. A role of HLA genes was investigated in cervical cancer. HLA-C group 1 was significantly more transmitted with invasive cervical cancer [60]. *PRKAG2* encodes a subunit of AMP-activated protein kinase (AMPK), which is a cellular homeostasis sensor. Sakabe et al. reported that *PRKAG2* expression was correlated with survival in liver cancer patients with IFN-a/5-FU treatment [46]. Similar results were observed in liver and lung cancer patients [61,62]. *TMEM108* may have a role in IFN signaling through the Wnt-b-catenin pathway. eQTL analyses suggested that the mechanism of *THEM108* is through the immune infiltrating cells and adjacent non-involved tissue around the tumor [45].

This study has strengths and limitations. The first strength is that it is the largest study of epigenome-wide DNA methylation profiles of PR PCa patients. We previously reported results of epigenome-wide methylation analysis based on 24 PR PCa cases [26]. The second strength is that the identification of DNA methylation profiles associated with risk and aggressiveness of PCa in PR H/L men will provide a potential mechanism for studying PCa disparities in the PR population and to generate hypotheses for future studies. Apprey et al. (2019) reported significant different methylation profiles between racial groups [18]. We are currently investigating whether the ancestry proportions influence the methylation patterns in PR H/L men with PCa. As limitations, the sample size of this study, especially for the aggressive PCa phenotype, was small. Therefore, this study lacked enough statistical power for broad generalizations. In addition, the lack of an independent validation set for confirmation of our findings is another limitation factor. In future, we plan to confirm our findings in cfDNA or DNA from urine or blood of PCa patients. This approach with minimal invasive methods may represent an important contribution in the clinical management of these patients.

In summary, we identified 141 differentially methylated genes in tumor tissues from PR PCa patients with the aggressive phenotype. Although our findings still need further validation, they provide an important insight into the epigenetic landscape of prostate cancer in the PR H/L patient population. Some of the genes identified in this study were associated with various cancers, including PCa, and affect various biological processes, such as immune pathways, cell signaling, metabolism, DNA repair, proliferation, and cell cycle.

## Figures and Tables

**Figure 1 biomolecules-12-00002-f001:**
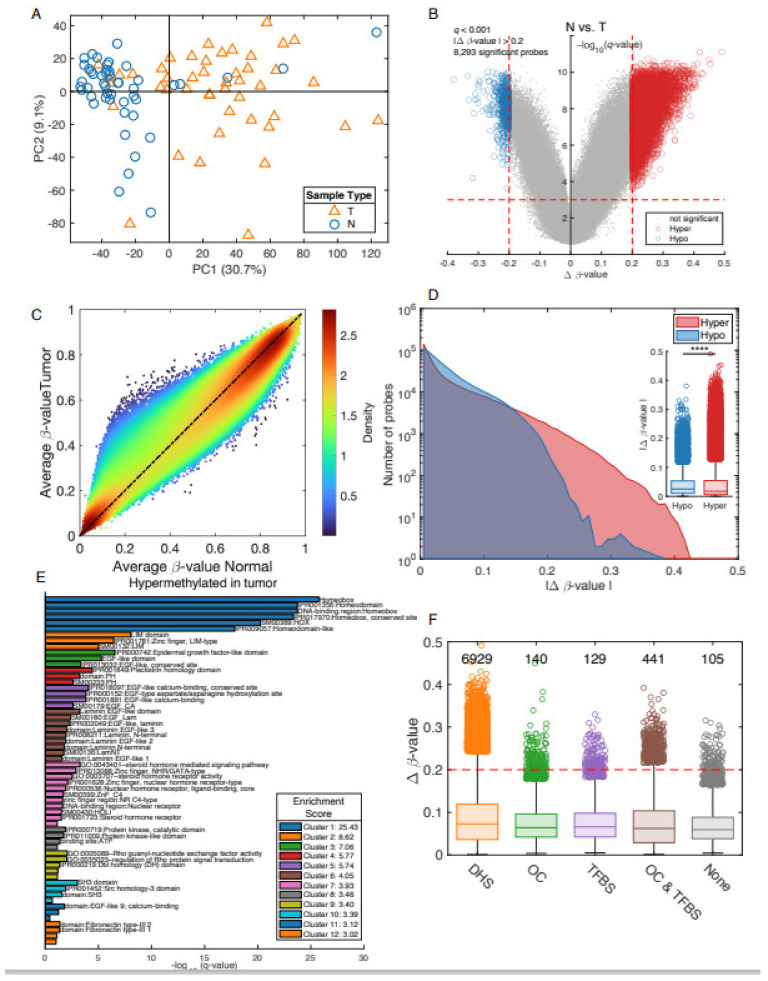
Analysis of tumor vs. normal tissues. The two first principal components in a PCA model using all 49 tumor and 49 adjacent normal samples and all CpG-probes (*n* = 807,583) separates normal (blue circles) and tumor (orange triangles) tissues from each other (**A**). A volcano plot comparing tumor vs. normal tissues with Δ*β*-value on the *x*-axis and FDR corrected *p*-value, −log_10_(*q*-value), on the *y*-axis (**B**). Blue circles indicate significant hypo-methylated CpG-probes and red circles hyper-methylated CpG-probes. Scatter density graph of average β-value for tumor samples vs. normal samples (**C**). Histogram comparing the number of hypo- and hyper-methylated probes (**D**). Enriched gene-sets for hyper-methylated probes using DAVID (**E**). Comparing the Δ*β*-value for significant hyper-methylated CpG-probes (**F**). DNase Hypersensitivity CpG-probes (DHS), Open Chromatin probes (OC), Transcription factor binding sites (TFBS). **** *p* < 0.0001.

**Figure 2 biomolecules-12-00002-f002:**
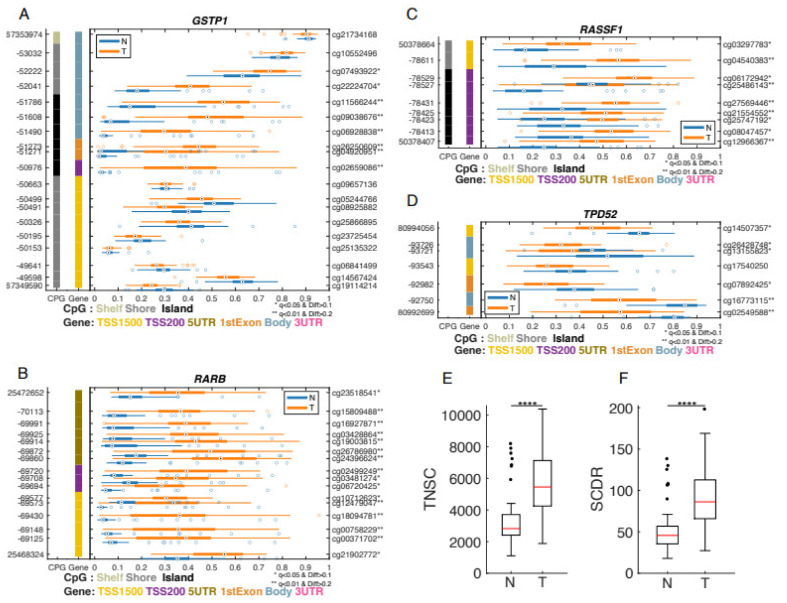
Differently methylated genes in tumor vs. normal tissues. GSM plots comparing the methylation levels between normal and tumor tissues for *GSTP1* (**A**), *RARB* (**B**), *RASSF1* (**C**), and *TPD52* (**D**). The selected CpG probes are shown on the *y*-axis with probe-id on the right *y*-axis and the genomic position on the left *y*-axis. The methylation level for each CpG probe is shown along the *x*-axis (β-value) with a boxplot for normal samples (N) in blue and tumor samples (T) in orange. The leftmost column indicated CpG island while the second column indicates the CpG probes location in the gene. Total number of stem cell divisions per stem cell (TNSC) boxplot comparing tumor vs. normal tissues (**E**), and stem cell division rate (SCDR) boxplot (**F**). **** *p* < 0.0001.

**Figure 3 biomolecules-12-00002-f003:**
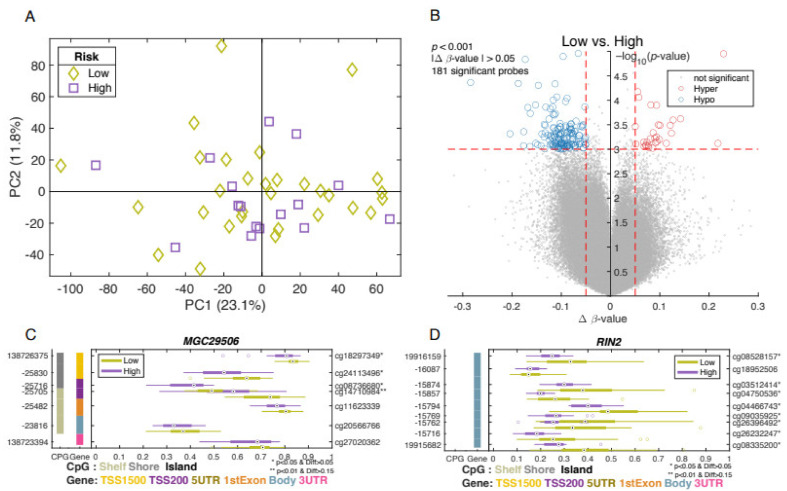
Analysis of high-risk vs. low risk. The two first principal components in a PCA model using all tumor samples (*n* = 49) and all CpG-probes (*n* = 785,071) with low-risk tumors (green diamonds) and high-risk tumors (purple squares) (**A**). A volcano plot comparing low-risk tumors vs. high-risk tumors with Δβ-value on the *x*-axis and *p*-value, −log_10_(*p*-value), on the *y*-axis (**B**). Blue circles indicate significant hypo-methylates CpG-probes and red circles hyper-methylated CpG-probes. GSM plots comparing the methylation levels between low-risk tumors and high-risk tumors for MGC29506 (**C**), RIN2 (**D**).

**Figure 4 biomolecules-12-00002-f004:**
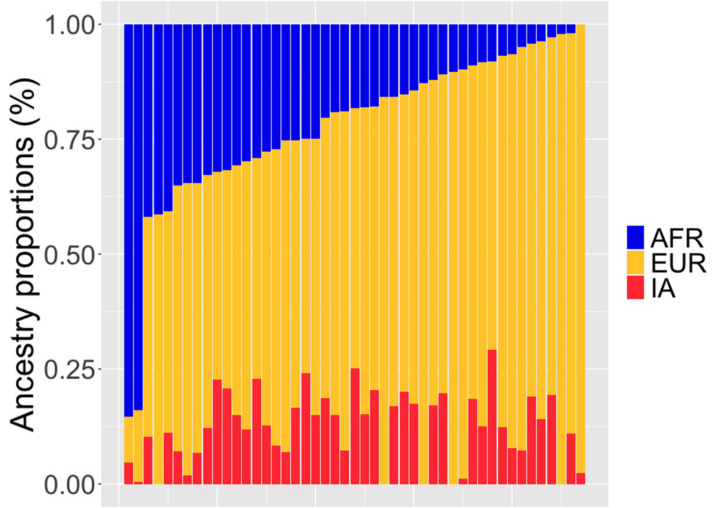
Visualization of the ancestry proportions for each individual in 49 PR H/L men with PCa (Puerto Rico PCa) compared to 1000 Genomes Admixed Americans populations. Assuming three ancestral populations (*k* = 3), each column represents one individual, and each color corresponds to the contribution of each ancestral population to the genome of a given individual (blue = African, yellow = European, and red = Indigenous American).

**Table 1 biomolecules-12-00002-t001:** Clinicopathological characteristics of Puerto Rican men (*n* = 49) with prostate cancer in the pilot study.

Risk	High	Low	*p*-Value
	*n* = 17	*n* = 32	
Age at Diagnosis	64.6 ± 5.7	60.3 ± 9.1	0.085
PSA	7.72 ± 5.07	7.01 ± 6.46	0.724
Gleason score			<0.0001
6	0	20	
7 (3 + 4)	0	12	
7 (4 + 3)	11	0	
8–9	6	0	
Stage			<0.0001
T1c	1	1	
T2a	2	9	
T2c	8	20	
T3a	1	1	
T3b	5	1	
Surgical margins			0.71
Yes	1	3	
No	14	27	
Missing	2	2	

*p*-values were obtained from Student’s t, chi-square, or Fisher exact tests. PSA, prostate-specific antigen.

**Table 2 biomolecules-12-00002-t002:** Ancestry proportions in the study cohort (*n* = 49).

Ancestral Population	Average	SD	Maximum	Minimum
African	0.219	0.178	0.853	0.00001
European	0.658	0.179	0.978	0.0995
Indigenous American	0.123	0.0784	0.292	0.00001

SD: standard deviation.

## Data Availability

The data presented in this study are available on request from the corresponding author. The data are not publicly available due to privacy of participants.

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
