# Peer review of "Dysregulation of DNA Methylation and Epigenetic Clocks in Prostate Cancer among Puerto Rican Men"

_biomolecules, 2021, doi:10.3390/biom12010002_

Round 1
Reviewer 1 Report
Berglund an the co-authors presented their findings on a DNA Methylation study using the EPIC array in a US-population suffering from prostate cancer.
The presented manuscript is sound in most of the parts. However, I find some contradictory and incomprehensible passages that need revision.
- the sample numbers in the Abstract and in the Methods part are misleading - either provide complete information about sample numbers in the Abstract, or eliminate it completely. in the Abstract you talk about 17 and 32 samples per group. on the next pages you have 96 samples... this is hard to read.
- there is no validation data included in the manuscript, although the authors talk about such an ongoing study. it is might be worth to wait until the validation data is ready and to include it in the present manuscript. this would make it more sound. However, this contradicts an "optimized" puplication strategy.
- chapter 2.2.3, line 149: please provide more information how you defina a DMR.
- 3.1, line 169/170: this sentence is incomplete
- 3.2, line 190: what does the symbol after the word "average" mean? this sympol appears from time to time in the manuscript. (e.g. also at the end of line 199)
- In general, the results part is very poor and I recommend extensive revision of this part. you have to underpin you statements with numbers. this was done partly in the discussion section. However, I do not like mixture of results/discussion in the discussion-section.
- the impact on biology of the gathered isn't analyzed properly. add a section to show the results on this, a section for this is also needed in the disucssion.
- section 4, line 308/309: this sentence is either not completed, or needs rewording
- section 4, line 309: you talk about the "long-term", add also an explanation of your vision and depict/describe in more detail how you can influence the current status in the future with your molecular tools.
- section 4, line 311: how many genes thit you identify?
- section 4, line 322-333: this should be moved to the results section.
- section 4, line 347-354: is it planned to confirm this findings cfDNA or DNA from urine? this would bring additional insight to your findings, and if this can be confirmed in samples collected by minimal invasive methods, also an additional major value.
- section 4, line 378-381: you need to elaborate on this findings in the results section and underpin your statements with the respective numbers.
Author Response
Reviewer 1
- the sample numbers in the Abstract and in the Methods part are misleading - either provide complete information about sample numbers in the Abstract, or eliminate it completely. in the Abstract you talk about 17 and 32 samples per group. on the next pages you have 96 samples... this is hard to read.
Response: The total number of Puerto Rican prostate cancer patients is 49. We used 49 pairs, thus 49 tumor and 49 adjacent normal tissues for DNA methylation analysis. These corrections are reflected in Lines 35, 188, 189, 272, and 294.
- there is no validation data included in the manuscript, although the authors talk about such an ongoing study. it is might be worth to wait until the validation data is ready and to include it in the present manuscript. this would make it more sound. However, this contradicts an "optimized" publication strategy.
Response: We completely agree with reviewer’s comment. It would be ideal to have an independent validation set to confirm our findings. However, the sample collection for such validation set will take additional 1-2 years. Therefore, we will report our findings first, and then evaluation will be done with newly collected samples as a validation set. The lack of a validation set was described as a limitation in the Discussion, lines 439-440.
- chapter 2.2.3, line 149: please provide more information how you define a DMR.
Response: We have now expanded this whole section with a better description of how both of Differentially Methylated Probes (DMPs) and Regions (DMRs) was determined lines 154-162.
- 1, line 169/170: this sentence is incomplete
Response: The sentence was revised “As expected, the significantly different distribution in clinical stage was detected between two groups.” lines 179-180
- 2, line 190: what does the symbol after the word "average" mean? this symbol appears from time to time in the manuscript. (e.g. also at the end of line 199)
Response: This should have been a beta (β) symbol, it appears to have been accidentally converted to another symbol in the submission process. We have now fixed this throughout the manuscript, lines 193, 197, 204, 205, 209, 247, 274, 277, and 279.
- In general, the results part is very poor and I recommend extensive revision of this part. you have to underpin you statements with numbers. this was done partly in the discussion section. However, I do not like mixture of results/discussion in the discussion-section.
Response: As the reviewer recommended, we have added more text describing the results in the Results section and moved text from the Discussion to the results section. Lines 219-230, 257-260.
- the impact on biology of the gathered isn't analyzed properly. add a section to show the results on this, a section for this is also needed in the discussion.
Response: We added paragraphs regarding biology in the Results and Discussion sections. line 257-260 in Results and 405-426 in Discussion.
- section 4, line 308/309: this sentence is either not completed, or needs rewording
Response: We revised the sentence as “We observed that several differentially methylated genes were found in aggressive tumor tissue from PR PCa patients.” Line 334
- section 4, line 309: you talk about the "long-term", add also an explanation of your vision and depict/describe in more detail how you can influence the current status in the future with your molecular tools.
Response: We described the potential impact of our molecular tools. line 335-346
- section 4, line 311: how many genes thit you identify?
Response: We have now added the number of genes that we identified. 892 genes were identified between tumor and normal tissues. Line 347
- section 4, line 322-333: this should be moved to the results section.
Response: As the reviewer requested, we moved this part to the Results section, line 219-230.
- section 4, line 347-354: is it planned to confirm this findings cfDNA or DNA from urine? this would bring additional insight to your findings, and if this can be confirmed in samples collected by minimal invasive methods, also an additional major value.
Response: We appreciate your valuable guidance. We added your suggestions as the future study in the Discussion section. Line 440-442
- section 4, line 378-381: you need to elaborate on this findings in the results section and underpin your statements with the respective numbers.
Response: We have now added the number of genes and re-written this section. Lines 443-450.
Reviewer 2 Report
It was a pleasure reviewing the manuscript "Dysregulation of DNA Methylation and Epigenetic Clocks in Prostate Cancer among Puerto Rican Men" by Berglund et al.
The authors used ADMIXTURE for global ancestry proportions and Illumina for methylation patterns. They identified several methylation patterns which after external validation may help explain prostate cancer aggressiveness in Peurto Rico men. Overall the article is well written. I do not have any specific questions.
Author Response
It was a pleasure reviewing the manuscript "Dysregulation of DNA Methylation and Epigenetic Clocks in Prostate Cancer among Puerto Rican Men" by Berglund et al.
- The authors used ADMIXTURE for global ancestry proportions and Illumina for methylation patterns. They identified several methylation patterns which after external validation may help explain prostate cancer aggressiveness in Puerto Rico men. Overall the article is well written. I do not have any specific questions.
Response: We appreciate reviewer’s kind words.
Reviewer 3 Report
This study was reported the association between dysregulation of DNA methylation and the aggressive behavior of prostate cancer in Puerto Rican patients. The reviewer would like to suggest some critiques as follows.
- On line 57, what is “an over-generalization regarding variations in PCSM rates”?
- On line 68, what is poor outcome? Oncological outcomes?
- On line 81, what is PCa risk?
- On line 84, what is “To maximize statistical efficiency”?
- With regard to epigenetic differences between prostate cancer and adjacent normal tissue, the aim of this study was “expand on the knowledge”? investigate?
- On line 377, “Although” is correct?
- On 379, what is “Other genes”?
Author Response
Reviewer 3.
This study was reported the association between dysregulation of DNA methylation and the aggressive behavior of prostate cancer in Puerto Rican patients. The reviewer would like to suggest some critiques as follows.
- On line 57, what is “an over-generalization regarding variations in PCSM rates”?
Response: We regret for not being clear. We meant all Hispanic/Latino should not combined into one group. We revised the sentence. Line 58-60
- On line 68, what is poor outcome? Oncological outcomes?
Response: As the reviewer guessed correctly, it is bad disease outcome. We added examples of poor outcome of prostate cancer in the sentence. Line 70
- On line 81, what is PCa risk?
Response: PCa risk meant “chance to get prostate cancer”. We revised the sentence.
- On line 84, what is “To maximize statistical efficiency”?
Response: If many genes were investigated with a small sample size, most results will not be statistically significant, or higher change of false positive. We revised the sentence. Line 84-86
- With regard to epigenetic differences between prostate cancer and adjacent normal tissue, the aim of this study was “expand on the knowledge”? investigate?
Response: For clarification of our goal, we revised the sentence below.
“Our findings on epigenetic differences between prostate tumor and adjacent non-involved (normal) tissues will be used to detect PCa early and may provide how PCa starts from the normal tissues.” Line 114-116
- On line 377, “Although” is correct?
Response: Thanks for pointing out. it should be below. “Although our findings still need further validation, they provide an important insight into the epigenetic landscape of prostate cancer in the PR H/L patient population” line 444-446
- On 379, what is “Other genes”?
Response: We have now re-written part of the conclusion to be more clear
Some of the genes identified in this study were associated with various cancers including PCa, and affect various biological processes, such as immune pathways, cell signaling, metabolism, DNA repair, proliferation, and cell cycle. Line 448-449
Round 2
Reviewer 3 Report
The authors revised the paper in accordance with the reviewers’ comments. The reviewer believes that this paper will provide useful information for readers.